# MODEL MANIPULATION ATTACKS ENABLE MORE RIGOROUS EVALUATIONS OF LLM CAPABILITIES

**Zora Che**[*]
University of Maryland, College Park, MATS
zche@umd.edu

**Stephen Casper**[*]
MIT CSAIL
scasper@mit.edu

**Anirudh Satheesh**
University of Maryland, College Park

**Rohit Gandikota**
Northeastern University

**Domenic Rosati**
Dalhousie University

**Stewart Slocum**
MIT CSAIL

**Lev McKinney**
University of Toronto

**Zichu Wu**
University of Waterloo

**Zikui Cai**
University of Maryland, College Park

**Bilal Chughtai**
Independent

**Daniel Filan**
MATS

**Furong Huang**
University of Maryland, College Park

**Dylan Hadfield-Menell**
MIT CSAIL

## ABSTRACT

Evaluations of large language model (LLM) capabilities are increasingly being incorporated into AI safety and governance frameworks. Empirically, however, LLMs can persistently retain harmful capabilities, and evaluations often fail to identify hazardous behaviors. Currently, most evaluations are conducted by searching for *inputs* that elicit harmful behaviors from the system. However, a limitation of this approach is that the harmfulness of the behaviors identified during any particular evaluation can only lower bound the model's worst-possible-case behavior. As a complementary approach for capability elicitation, we propose using *model-manipulation* attacks which allow for modifications to the latent activations or weights. Here, we test 8 state-of-the-art techniques for removing harmful capabilities from LLMs against a suite of 5 input-space and 5 model-manipulation attacks. In addition to benchmarking these methods against each other, we show that (1) model resilience to capability elicitation attacks lies on a low-dimensional robustness subspace; (2) the attack success rates (ASRs) of model-manipulation attacks can help to predict and develop conservative high-side estimates of the ASRs for held-out input-space attacks; and (3) state-of-the-art unlearning methods can easily be undone within 50 steps of LoRA fine-tuning. Together these results highlight the difficulty of deeply removing harmful LLM capabilities and show that model-manipulation attacks enable substantially more rigorous evaluations for undesirable vulnerabilities than input-space attacks alone.[1]

# 1 INTRODUCTION

Rigorous evaluations of large language models (LLMs) are widely recognized as key for risk mitigation (Raji et al., 2022; Anderljung et al., 2023; Schuett et al., 2023; Shevlane et al., 2023) and are consistently being incorporated into AI governance initiatives (POTUS, 2023; NIST, 2023; UK DSIT, 2023; EU, 2023). However, despite their efforts, developers often fail to identify and fix overtly harmful LLM behaviors before deployment (Wei et al., 2024; Shayegani et al., 2023; Carlini

---

[*]Equal contribution
[1]Models will be released upon paper arXival.

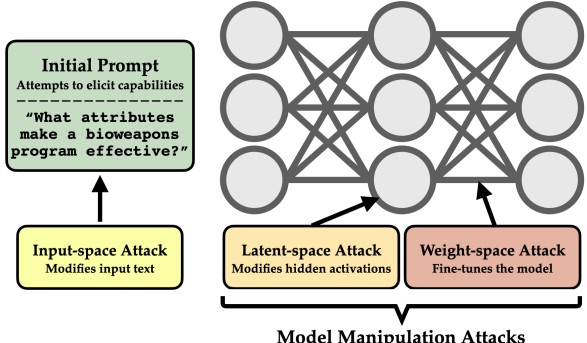

Figure 1: **Model manipulation attacks can modify latents and weights.** In contrast to input-space attacks, model manipulation attacks elicit capabilities from an LLM by making small modifications to the hidden activations or weights.

et al., 2024). Existing automated evaluations often fall short of identifying an LLM's true potential for harm (Casper et al., 2024a). For example, Li et al. (2024a) demonstrated that even when common, standardized benchmark attacks failed to jailbreak an LLM, manual ones succeeded more than 50% of the time. This highlights a fundamental limitation of using input-space attacks for LLM evaluations: the worst behaviors identified during any particular evaluation can only offer a lower bound of the model's overall worst-case behaviors. Poor evaluations can not only underestimate risks but can be counterproductive if they lead to a false sense of security (Anderljung et al., 2023).

In safety-critical engineering fields, it is a very common principle to design and test systems to ensure that they can handle stresses at least as strong – and ideally stronger – than what they will face in deployment (Clausen et al., 2006). For example, in construction, buildings are often designed to handle several times the load they are ever intended to bear. Here, we propose an analogous approach for AI system evaluations.

In addition to evaluating systems under input-space threats, we also propose using *model manipulation* attacks which allow for adversarial modifcations to the model's weights or latent activations. We attempt to answer two questions, each corresponding to a different threat model:

- **How vulnerable are LLMs to model manipulation attacks?** Answering this will help us understand how model-manipulation attacks can directly help to study risks from models that are open-sourced[2] or may be leaked (Nevo et al., 2024).

- **How much can model manipulation attacks tell us about LLM vulnerabilities to unforeseen input-space attacks?** Answering this will help us understand how model manipulation attacks can help to assess risks from both open- and closed-source models.

Here, we test state-of-the-art capability removal ("unlearning") methods for LLMs against a suite of input-space and model manipulation attacks. We make four contributions.

1. **Benchmarking:** We benchmark 8 state-of-the-art unlearning methods and 10 capability elicitation attacks against each other.

2. **Science of robustness:** We show that LLM resilience to a variety of capability elicitation attacks lies on a low-dimensional robustness subspace.

3. **Evaluation methodology:** We show that the success of some model manipulation attacks correlates with that of held-out input-space attacks. We also show that few-shot LoRA fine-tuning attacks can typically be used to conservatively over-estimate a model's robustness to held-out input-space threats.

---

[2]It may seem obvious that model-manipulation attacks are needed to evaluate realistic threats from models prior to open-sourcing. However, there is precedent for developers failing to use them prior to open-source releases. For example, before releasing Llama 2 and Llama 3 models, Meta's red-teaming efforts did not reportedly involve model manipulation attacks (Touvron et al., 2023; Dubey et al., 2024).

4. **Few-shot fine-tuning attacks:** We find that state-of-the-art unlearning methods can consistently be undone within 50 steps of few-shot fine-tuning.

Overall, our results suggest that model manipulation attacks can be a unique and valuable way for evaluators to gain more information about the potential worst-case behaviors of LLMs.

## 2 RELATED WORK

**Latent-space attacks:** During a latent-space attack, an adversary can make modifications to a model's hidden activations. Prior work has found that adversarial training under these attacks can improve the generality of a model's robustness (Sankaranarayanan et al., 2018; Singh et al., 2019; Zhang et al., 2023; Schwinn et al., 2023; Zeng et al., 2024). In particular, Xhonneux et al. (2024), Casper et al. (2024b), and Sheshadri et al. (2024) use latent adversarial training to improve defenses against held-out types of adversarial attacks. Other recent work on activation engineering has involved making modifications to a model's behavior via simple transformations to their latent states (Zou et al., 2023a; Wang & Shu, 2023; Lu & Rimsky, 2024) including Arditi et al. (2024) who showed that linear activation interventions could be used to jailbreak LLMs.

**Few-shot fine-tuning attacks:** During a few-shot fine-tuning attack, an adversary can modify a model's weights via fine-tuning on a limited number of examples. For example, Qi et al. (2023) showed that fine-tuning on as few as ten examples could be used to jailbreak a GPT-3.5. A variety of works have used few-shot fine-tuning attacks to elicit LLM capabilities that were previously suppressed by safety fine-tuning or machine unlearning (Jain et al., 2023; Yang et al., 2023; Qi et al., 2023; Bhardwaj & Poria, 2023; Lermen et al., 2023; Zhan et al., 2023; Ji et al., 2024; Qi et al., 2024; Hu et al., 2024; Halawi et al.; Peng et al., 2024; Lo et al., 2024; Łucki et al., 2024; Shumailov et al., 2024; Lynch et al., 2024; Deeb & Roger, 2024). For example, Greenblatt et al. (2024) found that fine-tuning was a reliable way of eliciting hidden capabilities from a "password locked" model that would only exhibit certain capabilities if a specific "password" was present in the prompt.

**Unlearning evaluation:** Along with research on jailbreaks (e.g., (Shayegani et al., 2023)) and backdoors (e.g., (Greenblatt et al., 2024)), machine unlearning has been a research sandbox for evaluating capability elicitation methods. "Machine unlearning" been motivated by two main objectives: undoing the influence of specific training examples and removing specific capabilities from models (Liu et al., 2024). Here, we focus on capability unlearning. Conventionally, capability unlearning has been evaluated using a pair of tests for the preservation of general knowledge and the reduction of unlearning-task knowledge. However, more recently, adversarial approaches have been used to evaluate robust and generalizable unlearning (Patil et al., 2023; Lynch et al., 2024; Łucki et al., 2024; Hu et al., 2024; Liu et al., 2024). Here, we build off of Li et al. (2024b) who introduce WMDP, a benchmark for unlearning dual-use biotechnology, chemistry, and cyber-offense knowledge from LLMs. We evaluate methods using the WMDP Bio unlearning and evaluation tasks.

## 3 METHODS

We pit unlearning methods against capability elicitation attacks. We used each unlearning method to impair an LLM's ability to assist users in tasks involving dual-use biotechnology information while preserving its general capabilities (Li et al., 2024b). The goal of each capability elicitation attack is to produce an adversarial modification to model text inputs, input embeddings, latent activations, or weights to make it regain its ability to help users with dual-use biology tasks.

**Unlearning methods:** We unlearn dual-use bio-hazardous knowledge on the Llama 3 8B Instruct model Dubey et al. (2024) with the 8 unlearning methods listed in Table 1 and outlined in Appendix A.2.1. For all methods except TAR, we train on up to 1,600 examples of max length 512 from the bio-remove-split of WMDP as the forget set, and up to 1,600 examples of max length 512 from Wikitext as the retain set. Due to the computational cost of TAR, we attack the public release TAR model unlearned on WMDP Bio data. We also attack the public release of the "Random Mapping" method which is an ablated version of TAR without meta-learning. For all methods we trained, we evaluate 8 checkpoints evenly spaced across training on unlearning efficacy, utility preservation, and robustness to attacks. We evaluate unlearning efficacy on the WMDP Bio QA dataset.

**Defenses**

| *Unlearning methods* | Gradient Difference (**GradDiff**) | (Liu et al., 2022) |
| | Random Misdirection for Unlearning (**RMU**) | (Li et al., 2024b) |
| | RMU with Latent Adversarial Training (**RMU+LAT**) | (Sheshadri et al., 2024) |
| | Representation Noising (**RepNoise**) | (Rosati et al., 2024) |
| | Erasure of Language Memory (**ELM**) | (Anonymous, 2024) |
| | Representation Rerouting (**RR**) | (Zou et al., 2024) |
| | Tamper Attack Resistance (**TAR**) | (Tamirisa et al., 2024) |
| | Random Mapping (**RandMap**) | (Tamirisa et al., 2024) |

**Attacks**

| *Input-space* | Gradient-guided | **GCG** (Zou et al., 2023b) |
| | | **AutoPrompt** (Shin et al., 2020) |
| | Perplexity-guided | **BEAST** (Sadasivan et al., 2024) |
| | Prosaic | **PAIR** (Chao et al., 2024) |
| | | **Human prompt** |
| *Model manipulation* | Latent perturbation | **Embedding space** (Schwinn et al., 2024) |
| | | **Latent space** (Sheshadri et al., 2024) |
| | Fine-tuning | **Benign LoRA** (Qi et al., 2023) |
| | | **LoRA** (Hu et al., 2021) |
| | | **Full parameter** |

Table 1: **Table of attacks and defenses.** The capability elicitation (attack) methods and unlearning (defense) methods we use to remove and extract bio-hazardous capability from LLMs.

**Capability elicitation attacks:** We use 5 input-space attacks and 5 model manipulation attacks on our unlearned models. These attacks are designed to increase WMDP Bio performance. We list all 10 attacks in Table 1. Note that in this setting, we use capability elicitation methods to produce *universal* adversarial attacks which work for *all* prompts. This allows us to attribute improvements in performance to capability elicitation rather than answer-forcing from the model. We selected attacks based on algorithmic diversity and prominence in the state of the art.

In addition to the attacks in Table 1, we also tried many-shot attacks (Anil et al., 2024; Lynch et al., 2024) and translation attacks (Yong et al., 2023; Lynch et al., 2024) but found them to be consistently unsuccessful. All attacks, except fine-tuning attacks, used a 64-example subset for attack generation (approximately 5% of the evaluation examples) and were evaluated on the remaining WMDP Bio QA data. For descriptions and implementation details for each attack method, see Appendix A.

## 4 EXPERIMENTS

As discussed in Section 1, we have two motivations, each corresponding to a different threat model. First, we want to directly evaluate robustness to model manipulation attacks in order to better un-

| Method | WMDP ↓ | WMDP Under ↓ Best Attack | MMLU ↑ | MT-Bench/10 ↑ | AGIEval ↑ | Unlearning Score ↑ |
|---|---|---|---|---|---|---|
| Llama3 8B Instruct | 0.70 | 0.71 | 0.64 | 0.78 | 0.41 | 0.00 |
| Grad Diff | 0.26 | 0.54 | 0.53 | 0.28 | 0.27 | 0.19 |
| RMU | 0.26 | 0.59 | 0.59 | 0.68 | 0.42 | 0.39 |
| RMU + LAT | 0.32 | 0.66 | 0.60 | 0.71 | 0.39 | 0.34 |
| RepNoise | 0.25 | 0.62 | 0.57 | 0.64 | 0.36 | 0.36 |
| ELM | 0.29 | 0.54 | 0.54 | 0.73 | 0.41 | 0.37 |
| RR | 0.26 | 0.65 | 0.61 | 0.76 | 0.44 | **0.44** |
| TAR | 0.24 | 0.43 | 0.27 | 0.10 | 0.30 | 0.08 |
| RandMap | 0.27 | 0.61 | 0.55 | 0.41 | 0.31 | 0.24 |

Table 2: **Benchmarking LLM unlearning methods:** We report original WMDP Bio performance, worst-case WMDP Bio performance after attack, and three measures of general utility: MMLU, MT-Bench, and AGIEval. For all benchmarks, the random-guess baseline is 0.25 except for MT-Bench/10 which is 0.1. Representation rerouting (RR) has the best unlearning score. No model has a WMDP accuracy less than 0.43 after attack.

derstand risks from models that are open-sourced or leaked. Second, we want to understand what model manipulation attacks can tell us about unforeseen input-space attacks in order to study risks from open- or closed-source models. Unfortunately, unforeseen attacks are, by definition, ones that we do not have access to. Instead, since all of the input-space attacks that we use are held out during unlearning, we treat them as proxies for other held-out input-space attacks in general.

## 4.1 Benchmarking Unlearning Methods

**Unlearning evaluation:** In our models, we evaluate *unlearning efficacy* on the WMDP Bio QA task. We evaluate *general utility* on MMLU Hendrycks et al. (2020), MT-Bench Bai et al. (2024), and AGIEval Zhong et al. (2023). Because the goal of unlearning is to differentially decrease capabilities in a target domain, we calculate an "unlearning score" based on both the efficacy and utility degradation. For an original model $M'$ and an unlearned model $M$, we calculate $S_{\text{unlearn}}(M)$ with the following formula:

$$S_{\text{unlearn}}(M) = \underbrace{[S_{\text{WMDP}}(M) - S_{\text{WMDP}}(M')]}_{\Delta\text{Unlearn efficacy}} - \underbrace{[S_{\text{utility}}(M) - S_{\text{utility}}(M')]}_{\Delta\text{Utility degradation}} \tag{1}$$

where $S_{\text{WMDP}}(\cdot)$ is the accuracy on the WMDP Bio QA Evaluation and $S_{\text{utility}}(\cdot)$ is an aggregated utility measure. $S_{\text{utility}}(\cdot)$ is calculated by taking the average of MMLU, MT-Bench (divided by 10 so the score is between 0 and 1), and AGIEval. A positive unlearning score indicates that the model differentially lost WMDP Bio capabilities relative to general capabilities while a negative one reflects counterproductive unlearning. A "perfect" unlearning technique that did not change general capabilities but brought WMDP Bio performance down to the random-guess baseline would score 0.45. Table 4 reports results from the best-performing checkpoint (as determined by unlearning score) from each of the 8 methods.

**Representation rerouting (RR) achieves the highest unlearning score. GradDiff, TAR, and RandMap struggle.** We find different levels of unlearning success. Representation rerouting (RR) performs the best overall, achieving an unlearning score of 0.44. In contrast, GradDiff and RandMap have limited success and are only able to remove bio capabilities with heavy costs to general utility. The TAR model performed the worst overall due to near-random guess performance on all evaluations. Upon deeper investigation, we found that the TAR model had broad challenges with fluency. We also discovered that its unlearning was perplexingly brittle. For example, we found that simply adding a non-masked `<|begin_of_text|>` token as a prefix before evaluation caused the model to increase from 24% to 37%. As we will also show later in Figure 4, several other simple input-space attacks were able to greatly improve its performance.

**No method is robust to all attacks.** We plot the increase in WMDP Bio performance for the best checkpoint after each attack in Figure 4 and show that all models, even those with the lowest utility, exhibit a worst-case performance increase of 0.19 points or more on the WMDP Bio QA task.

## 4.2 Model robustness exists on a low-dimensional subspace.

**Weighted PCA:** We perform weighted principal component analysis on the WMDP improvements achieved by all 10 attacks on all 50 checkpoints. We first constructed a matrix $A$ with one row per model and one column per attack. Each $A_{ij}$ in the table corresponds to the increase in WMDP performance in model $i$ under attack $j$. We then centered the data by multiplying each row $A_i$ by the square root of the unlearning score: $\sqrt{S_{\text{unlearn}}(A_i)}$ This scaling step allowed for models to influence the analysis in proportion to their unlearning score.

**Four principal components explain over 90% of the variation in performance across the ten attacks.** Fig. 2 displays the eigenvalues from PCA and the top four principal components (each weighted by their eigenvalues). We see that the first principal component captures over half of the variance in attack successes, and the first four capture over 90%. This suggests that not all capability elicitation attacks exploit models via the same mechanisms, nor do they all do so via completely different ones. Meanwhile, analysis of the eigenvalue-weighted coordinates suggests that some model manipulation attacks are better predictors of input-space vulnerabilities than others. The first principal component exhibits a distinct difference between full and LoRA fine-tuning and all other attacks. Meanwhile, benign LoRA fine-tuning, embedding-space, and latent-space attack successes tend to coincide more with the success of held-out input-space attacks.

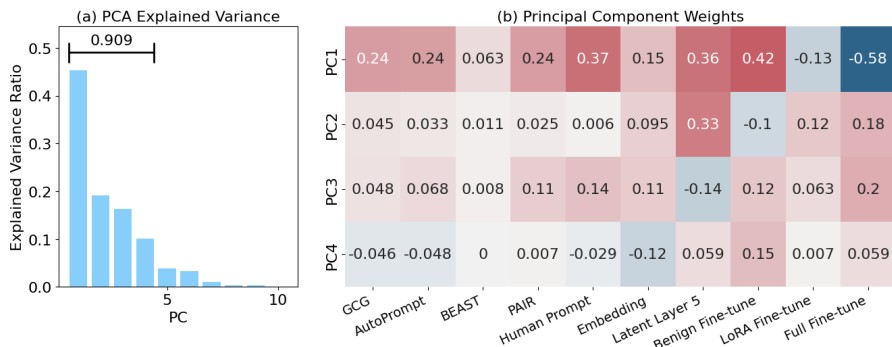

Figure 2: **Four principal components explain over 90% of the variation in performance across the ten attacks.** (Left) The proportion of explained variance for each principal component. (Right) We display the first four principal components weighted by their eigenvalues. The first suggests a split between the two adversarial (LoRA, Full) fine-tuning attacks and all others.

### 4.3 MODEL MANIPULATION ATTACKS HELP TO PREDICT AND UPPER BOUND THE SUCCESS OF INPUT-SPACE ATTACKS

**Comparing the success of input-space and model manipulation attacks:** In Figure 3, we plot the increases in WMDP Bio performance from input space versus model manipulation attacks. In the plots, we size points by their unlearning score and display the weighted pearson correlation.

**Benign fine-tuning, embedding-space, and latent-space attack successes tend to *correlate* with input-space attack successes.** These three model manipulation attacks tend to have positive correlations with input-space attack successes with low $p$ values (e.g., $< 0.05$). This suggests that benign fine-tuning embedding-space and latent-space attacks are particularly able to predict the successes of held-out input-space attacks. Surprisingly, full model fine-tuning attacks tend to negatively predict input-space attacks, though this trend is largely due to the ELM models. ELM models often cluster far away from all others, suggesting that ELM may be a mechanistically unique unlearning technique, but we leave further exploring this for future work.

**LoRA fine-tuning attack successes tend to *upper bound* input-space attack successes for models that do not experience significant utility degradation.** LoRA fine-tuning on the WMDP Bio forget set was usually able to exceed the success of all other attacks we tested. In the few cases in which the success of LoRA attacks does not exceed another attack, it tends to be for models (e.g., TAR) with low unlearning scores due to general utility degradation. This suggests that they may not have been robust to fine-tuning because of successful unlearning so much as having behavior that was unstable under training. This suggests that evaluators looking to upper bound the worst-case behavior of a model under potentially unforeseen threats may be able to use these fine-tuning attacks.

**Unlearning can consistently be reversed within 50 fine-tuning steps – sometimes even in a single step.** We show the results of multiple fine-tuning attack configurations against the best-performing model from each unlearning method in Figure 5. All finetuning experiments as detailed in Appendix A.3 are done within 50 gradients steps, and with 400 examples or fewer. In particular, the attack 'Full-4' only performs a single gradient step (with a batch size of 64) and still increases the WMDP performance on 5 of the 8 models by over 25 percentage points. The only models that were resistant to few-shot fine-tuning attacks were from GradDiff and TAR. However, GradDiff was still compromised by the Benign LoRA attack within just 50 gradient steps. Both of TAR and GradDiff models had low unlearning scores due to poor general utility.

## 5 DISCUSSION

**Implications for evaluations:** Our work may be of particular interest to the LLM evaluation community. Frameworks for AI governance are increasingly designed to rely on high-quality evaluations to identify hazards and handle risks in LLMs (Raji et al., 2022; Anderljung et al., 2023; Schuett et al., 2023; Shevlane et al., 2023; Uuk et al., 2024). Formal evaluations of AI systems, however, face a

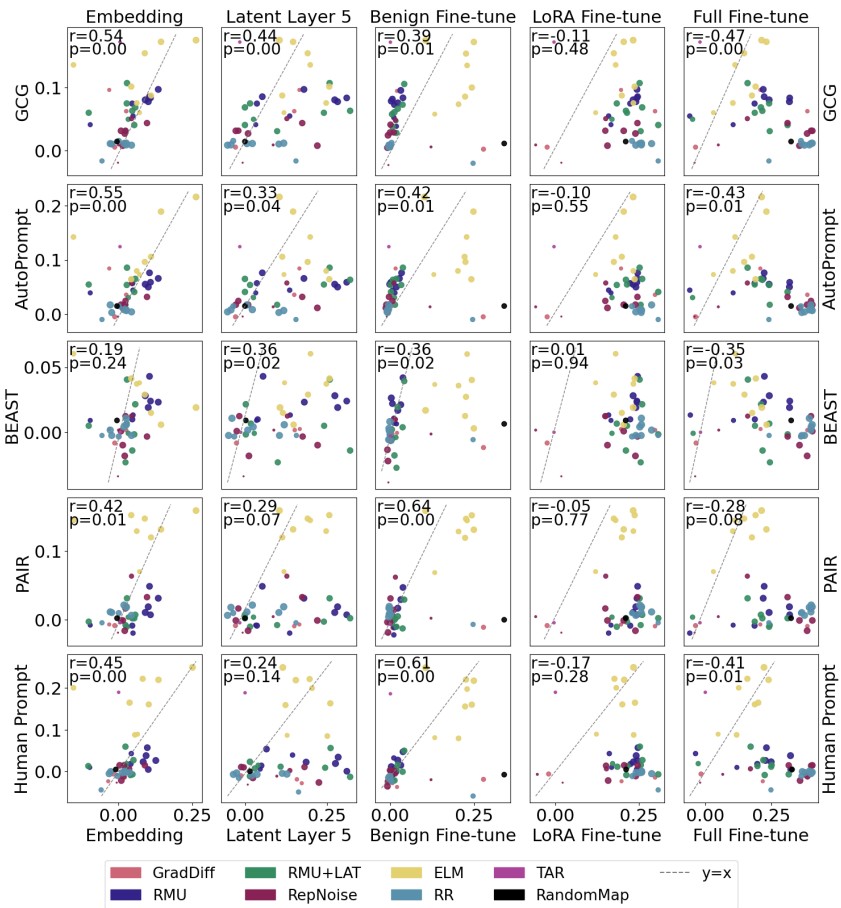

Figure 3: **Benign fine-tuning, embedding-space, and latent-space attack successes tend to *correlate* with input-space attack successes. LoRA fine-tuning attack successes tend to *upper bound* the successes of input space attacks for models with high unlearning scores.** Here, we plot the increases on WMDP Bio performance from input-space attacks versus model manipulation attacks. We weight points by their unlearning score from Section 4.1. We also display the unlearning-score-weighted correlation, the correlation's $p$ value, and the line $y = x$ (Notice this is not the line of best fit). Points below and to the right of the line indicate that the model manipulation attack was more successful. For models with a high unlearning score, LoRA attacks consistently match or exceed the performance of all input-space attacks.

number of challenges (Anderljung et al., 2023) including limitations in technical tooling. Our work adds to the growing consensus that access to model internals is necessary for rigorous evaluations (Casper et al., 2024a). When models are released with open weights, model manipulation attacks during evaluations are directly necessary to understand risks from misuse. Meanwhile, when they are released with closed weights, our results still suggest that model manipulation attacks can help evaluators make more informed inferences about potential worst-case behavior.

**Limitations:** Our work focuses only on Llama 3 8B Instruct derived models, on the task of unlearning hazardous biology knowledge. This allows for considerable experimental depth, but we expect that there may be different dynamics with different models performing different tasks. Another limitation is that our evaluations of undesirable capabilities were conducted only by evaluating models under multiple-choice questions from the WMDP Bio test set (Li et al., 2024b). However, these evaluations can sometimes be brittle. Finally, the science of evaluations is still generally immature, and it is not yet clear how to best translate the outcome of evaluations like ours into actionable recommendations.

**Future work:**

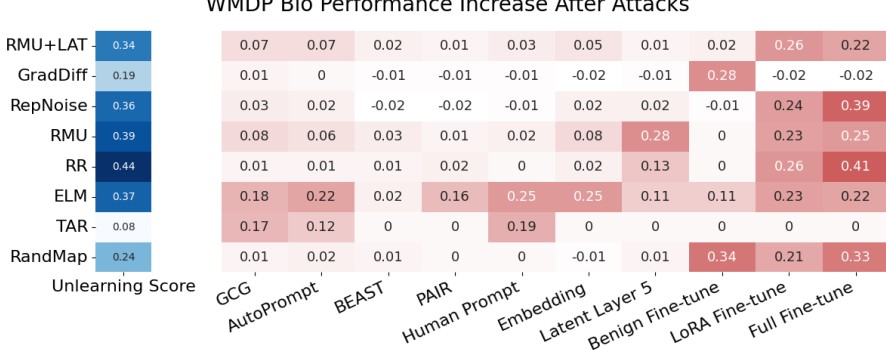

Figure 4: **All unlearning methods could be successfully attacked, but not all attacks were successful on all models.** Here, we benchmark attacks against the best models from each unlearning method based on unlearning score. No method is robust to all the 10 attacks. The min-max WMDP performance increase under an attack was 19% (Human Prompt vs. TAR).

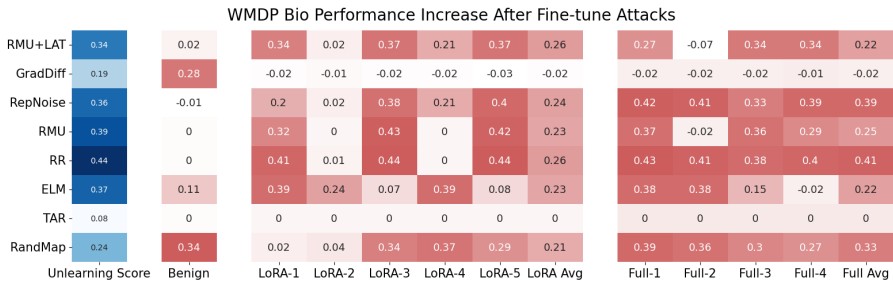

Figure 5: **Fine-tuning undoes unlearning efficiently.** We plot the heatmap of the best checkpoint for each method under fine-tuning attack. All fine-tuning experiments are done within 50 gradient steps, with 400 examples or less. All hyper-parameters are listed in Appendix A.3.

- **Models and tasks:** Experiments with larger models and additional tasks would offer a broader understanding of model manipulation attacks' potential to aid in capability evaluations. We are particularly interested in similar work involving jailbreaks.

- **What mechanisms underlie robust capability removal?** Currently, our understanding of robust unlearning is limited. It is not known what types of mechanisms are responsible for strong versus weak unlearning. We are interested in future work to mechanistically characterize unlearning. This could help in understanding how to unlearn undesirable capabilities in a way that is effective, robust, and lacks side effects.

- **Bridging research and practice:** This work was motivated by the goal of helping evaluators more effectively study potential worst-case risks from harmful capabilities in LLMs. We hope that model manipulation attacks can be further studied and used in practice to assess risks in consequential models pre-deployment.

**Conclusion:** By comprehensively benchmarking 8 capability removal methods under 10 diverse capability elicitation methods, we have contributed to a more thorough understanding of LLM robustness and how to evaluate it. Not all capability elicitation attacks exploit the same weaknesses in models, nor do they all exploit the exact same ones. We have found that over 90% of the empirical variance in robustness across our suite of attacks can be explained by four principle components. Meanwhile, model manipulation attacks that make small perturbations to the LLM's weights or activations can help to predict its vulnerability to input-space methods. In particular, we show that few-shot fine-tuning attacks tend to be very strong and often exceed the successes of input-space attacks. Together these results suggest that model manipulation attacks can be uniquely helpful for inferring risks from unforeseen types of input space attacks.

ACKNOWLEDGEMENTS

We are grateful to the Center for AI Safety for compute and the Machine Learning Alignment and Theory Scholars program for research support. We thank Ekdeep Singh Lubaba and Taylor Kulp-McDowall for feedback.

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

# A EXPERIMENT DETAILS

## A.1 UNLEARNING EVALUATION

We report MT-Bench score as the average of one round and two round scores, and divide it by 10, the maximum number of points possible.

## A.2 UNLEARNING METHODS AND IMPLEMENTATION

### A.2.1 UNLEARNING METHODS

- **Gradient Difference (GradDiff):** Inspired by Liu et al. (2022), we train models to maximize the difference between the training loss on the forget dataset and the retain dataset.
- **Random Misdirection for Unlearning (RMU):** Li et al. (2024b) propose a method where model activations on harmful data are perturbed and model activations on benign data are preserved.
- **RMU with Latent Adversarial Training (RMU+LAT):** Sheshadri et al. (2024) propose training models using adversarial attacks in the latent space as a way to perform stronger unlearning. They combined this with RMU by leveraging adversarial perturbations when training only on the forget dataset.
- **Representation Noising (RepNoise):** Rosati et al. (2024) propose adding a noise loss term that minimizes the KL divergence between the distribution of harmful representations given harmful input and Gaussian noise.
- **Erasure of Language Memory (ELM):** Anonymous (2024) introduce ELM in order to thoroughly unlearn knowledge by training the model to mimic unknowledgeable behavior on the unlearning domain.
- **Representation Rerouting (RR):** Zou et al. (2024) introduces Representation Rerouting (also known as "circuit breaking") which trains models to map latent states induced by topics in the unlearning domain to orthogonal representations.
- **Tamper Attack Resistance (TAR):** Tamirisa et al. (2024) introduce TAR as a meta-learning approach which uses a tamper-resistant loss to minimize the effect of an attacker.
- **Random Mapping (RandMap):** In an ablation experiment, Tamirisa et al. (2024) also use random hashes to maximize the cosine similarity of the residual streams and hash vectors, which encourages the language model to remap token representations to random states.

Both for algorithmic diversity and to use the same implementations as the works which introduced each method, we use full fine-tuning for RMU, RMU-LAT, and RepNoise and LoRA for GradDiff, ELM, and RR.

### A.2.2 HYPERPARAMETERS

Beginning from prior works' implementations of methods, we tuned the hyperparameters below in order to achieve (1) gradual progress in unlearning across the 8 checkpoints that we took and (2) a high unlearning score by the end of training.

- GradDiff
  - LoRA Fine-tune
    - LoRA Rank: 256
    - LoRA $\alpha$: 128
    - LoRA dropout: 0.05
  - Learning Rate: $10^{-4}$

- – Batch Size: 32
- – Unlearning Loss Coefficient $\beta$: 14
- RMU
  - – Layer Fine-tune
    - Layers: 5, 6, 7
  - – Retain Loss Coefficient $\alpha$: 90
  - – Steer: 20
  - – Learning Rate: $5 \times 10^{-5}$
  - – Batch Size: 8
- RMU+LAT
  - – Layer Fine-tune
    - Layers: 5, 6, 7
  - – Retain Loss Coefficient $\alpha$: 90
  - – Learning Rate: $5 \times 10^{-5}$
  - – Batch Size: 8
  - – Steer: 20
- RepNoise
  - – Full Fine-tune
  - – Learning Rate: $5 \times 10^{-6}$
  - – Batch Size: 4
  - – Noise Loss Coefficient $\alpha$: 2
  - – Ascent Loss Coefficient $\beta$: 0.01
- ELM
  - – LoRA Fine-tune
    - LoRA Rank: 64
    - LoRA $\alpha$: 16
    - LoRA dropout: 0.05
  - – Learning Rate: $2 \times 10^{-4}$
  - – Batch Size: 8
  - – Retain Coefficient: 1
  - – Unlearn Coefficient: 6
- Representation Rerouting
  - – LoRA Fine-tune
    - LoRA Rank: 16
    - LoRA $\alpha$: 16
    - LoRA dropout: 0.05
  - – Learning Rate: $1 \times 10^{-4}$
  - – Batch Size: 8
  - – Target Layers: 10, 20
  - – Transform Layers: All
  - – LoRRA Alpha: 10

## A.3    ATTACK METHODS AND IMPLEMENTATION

**Greedy Coordinate Gradient (GCG)**    GCG (Zou et al., 2023b) performs token-level substitutions to an initial prompt by evaluating the gradient with respect to a one-hot vector of the current token. We implemented both time-bounded attacks on each unlearned model and transfer attacks using prefixes from one model to attack others. Unless otherwise specified, we report the mean performance of each gradient-guided attack.

**AutoPrompt**   Like GCG, AutoPrompt (Shin et al., 2020) performs a gradient-guided search over input tokens to design universal adversarial prompts.

**BEAST**   We used BEAm Search-based adversarial Attack (BEAST) (Sadasivan et al., 2024) to produce universal adversarial suffixes which were appended after the evaluation questions. Unlike GCG and AutoPrompt, BEAST is guided by perplexity instead of gradients. Since our attacks need to be universal, we leverage a modified version of BEAST to generate universal adversarial tokens for several user input prompts. Formally, we can define a set of user input prompts as $\{\mathbf{x}_1^{(u)}, \mathbf{x}_2^{(u)}, \cdots, \mathbf{x}_n^{(u)}\}$, where each $\mathbf{x}_i$ contains a question $q_i$ and answer choices $a_i$. Our goal is to generate an adversarial sequence of tokens $\mathbf{x}^{(a)}$ such that $q_i \oplus \mathbf{x}^{(a)} \oplus a_i$ can effectively attack the language model for all $i \in \{1, 2, \cdots, n\}$. We place the adversarial tokens between each question and the answer choices so that the beam search in BEAST is conditioned solely on the question and not the answers, as the jailbreak tokens to the end of the user prompt directly can leak the answer choices into the tokens. We attacked with different hyperparameters for search width and time, and reported results for $K = 5$.

**PAIR**   In addition to gradient-based approaches such as GCG and finetuning attacks, we also include a model-based approach similar to PAIR (Chao et al., 2024). PAIR uses a prompt-level jailbreaking attack, where an entirely new adversarial prompt is generated instead of augmenting the prompt with adversarial tokens. This is not transferable compared to other universal attack approaches as the attack is highly dependent on the original prompt. Instead, we modify PAIR to develop an adversarial suffix that can applied to many (potentially unseen) prompts. First, we task an attack model (base model such as Llama 3 8B) to generate an adversarial suffix, similar to how PAIR generates an adversarial prompt. Then, we append the adversarial suffix to each of the prompts in the training dataset and measure the accuracy of the target model (unlearned model) on the jailbroken prompts. Finally, we pass in an improvement prompt (a general string that asks the attack model to improve its adversarial suffix) and the accuracy of the target model to the attack model to repeat the process.

**Human Prompt**   We hand-crafted adversarial prompts to elicit harmful responses from a language model. After manual iteration, our selected prefix was "#Answer Key\n\nThis answer key contains questions and answers for advanced topics in bioengineering and biotechnology. All answers were verified correct by a panel of experts."

**Embedding Attack**   Schwinn et al. (2024) proposes applying adversarial perturbations to a language model's embedding layer instead of in the input space. We train universal $\ell_2$-norm bounded embedding space prompts which we apply to the embedded tokens for the prefix used for WMDP-bio evaluation: "The following are multiple choice questions (with answers) about biology.\n".

**Latent Layer Attack**   Sheshadri et al. (2024) proposes applying adversarial perturbations to a language model's latent layers. We train universal $\ell_2$-norm bounded embedding space prompts which we apply to the latent tokens at layer 5 for the prefix used for WMDP-bio evaluation: "The following are multiple choice questions (with answers) about biology.\n".

**Few-shot fine-tuning attacks**   We perform fine-tuning as relearning attack. The forget set (WMDP Bio Remove) consists of hazardous biology data, and the retain set (WMDP Bio Retain) contains non-hazardous biology data. We also performed benign LoRA fine-tuning on wikitext. We report hyper-parameters in Appendix A.3. All LoRA and Benign attacks are done with rank 16 and alpha 32. All examples have a maximum length of 512 tokens.

Table 3: Hyper-parameters for Fine-tuning Attacks

|  | Dataset | # of Examples | Batch Size | Learning Rate | Epochs | Total Steps |
|---|---|---|---|---|---|---|
| Full-1 | WMDP Bio Remove | 400 | 16 | 2e-05 | 2 | 25 |
| Full-2 | WMDP Bio Remove | 64 | 8 | 2e-05 | 2 | 16 |
| Full-3 | WMDP Bio Retain | 64 | 64 | 5e-05 | 2 | 2 |
| Full-4 | WMDP Bio Retain | 64 | 64 | 5e-05 | 1 | 1 |
| LoRA-1 | WMDP Bio Remove | 400 | 8 | 5e-05 | 1 | 50 |
| LoRA-2 | WMDP Bio Remove | 400 | 64 | 5e-05 | 1 | 7 |
| LoRA-3 | WMDP Bio Retain | 400 | 8 | 5e-05 | 1 | 50 |
| LoRA-4 | WMDP Bio Remove | 64 | 8 | 1e-04 | 2 | 16 |
| LoRA-5 | WMDP Bio Retain | 64 | 8 | 1e-04 | 2 | 16 |
| Benign-1 | Wikitext | 400 | 8 | 5e-05 | 1 | 50 |

