# OpenReview forum: "Model Manipulation Attacks Enable More Rigorous Evaluations of LLM Capabilities"
_NeurIPS.cc/2024/Workshop/SafeGenAi — SafeGenAi Poster_

### Official Review · Reviewer_LFaK · 2024-10-08
**Benchmark the impact of model manipulation against LLM Unlearning**

**Rating:** 8
**Confidence:** 3

**Review:**

This paper presents a comprehensive benchmark of 5 input-space and 5 model-manipulation attacks against 8 unlearning methods. The authors draw several interesting conclusions, notably that model manipulation attacks can be uniquely helpful for inferring risks from unforeseen types of input space attacks. Additionally, they find that model resilience to capability elicitation attacks lies on a low-dimensional robustness subspace.
Overall, this research direction is commendable as it analyzes model-manipulation attacks beyond traditional text prompt attacks. The study provides valuable insights into the vulnerabilities and strengths of various unlearning methods against different types of attacks.

cons:
It would be better if you could propose some novel model manipulation methods according to your findings as the future exploration.

---

### Official Review · Reviewer_sxyD · 2024-10-09
**strong approach to generalize capability evaluations**

**Rating:** 8
**Confidence:** 3

**Review:**

This paper has a novel and excellent approach to try to generalize model capability evaluations. Basically, the attacker is given stronger capabilities during evaluation then they would have in the real world, with the ability to modify model weights, in hopes of exposing capabilities and hidden behavior.

This is similar to what the larger labs do when they fine tune a model to optimize for a specific safety violation, to see if they can get it to exhibit bad behavior. And is a stronger evaluation than what all the AI safety institutes seem to be doing right now.

Machine unlearning techniques are not very good right now, so being able to undo them is not particularly surprising. But this is still a helpful study. Lora can do undo everything with 50 steps and in some cases with just one step.

Pros:
- highly useful and practical recommendation to increase the span of safety evaluations
- grounded in existing industry approaches
- significant experimental results

Cons
- machine unlearning not that difficult to undo right now

---

### Official Review · Reviewer_L1F6 · 2024-10-09
**A good evaluation of different unlearning methods in LLMs**

**Rating:** 6
**Confidence:** 4

**Review:**

# Summary

Explores evaluating LLMs in the context of unlearning via latent space / weight space attacks, complimenting more traditional prompt based approaches. While there might be limited novelty in the work, the evaluations are done well, and some nice new insights are presented.

# Merits

- Easy to follow
- I agree with the authors that latent/weight space attacks are an important and powerful attack method to consider, especially in the context of evaluating unlearnt information
- Novelty is a bit limited, as there were other works discussing unlearning via latent space attacks [1], and undoing alignment with fine-tuning [2] (which is quite similar to unlearning in the context of this paper), but I think this is OK -- it is nice to have all of these approaches unified and evaluated in one work, and there are more novel findings in the context of fine-tuning attacks
- I like the direction of using latent/weight space attacks (extremely powerful threat models) as a way to predict/upper-bound the success of prompt based attacks; this is a potentially much cheaper way to concretely evaluate models vs prompt-based attacks

# Limitations

- I think the $S_{unlearn}$ score has oddly defined bounds for use as a metric, generally it makes more sense for a metric to have maximal values at cleaner numbers (e.g. 1) rather than 0.45. I think what you're trying to quantify makes sense, but making it a bit more intuitive would be useful for users to understand your tables/figures better
- It would have been nice to see your benchmarks applied to other unlearning tasks, such as [3]


[1] Schwinn, Leo, et al. "*Soft prompt threats: Attacking safety alignment and unlearning in open-source llms through the embedding space.*" https://arxiv.org/abs/2402.09063 (NeurIPS 2024).

[2] Lermen, Simon, Charlie Rogers-Smith, and Jeffrey Ladish. "*Lora fine-tuning efficiently undoes safety training in llama 2-chat 70b.*" https://arxiv.org/abs/2310.20624 (2023).

[3] Maini, Pratyush, et al. "*Tofu: A task of fictitious unlearning for llms.*" https://arxiv.org/abs/2401.06121